# Parenchymal border macrophages regulate tau pathology and tau-mediated neurodegeneration

Antoine Drieu[1,5], Siling Du[1,5], Michal Kipnis[2,3,4], Megan E Bosch[2,3,4], Jasmin Herz[1,5], Choonghee Lee[2,3,4], Hong Jiang[2,3,4], Melissa Manis[2,3,4], Jason D Ulrich[2,3,4], Jonathan Kipnis[1,5], David M Holtzman[2,3,4], Maud Gratuze[2,3,4,6]

Parenchymal border macrophages (PBMs) reside close to the central nervous system parenchyma and regulate CSF flow dynamics. We recently demonstrated that PBMs provide a clearance pathway for amyloid-$\beta$ peptide, which accumulates in the brain in Alzheimer's disease (AD). Given the emerging role for PBMs in AD, we explored how tau pathology affects the CSF flow and the PBM populations in the PS19 mouse model of tau pathology. We demonstrated a reduction of CSF flow, and an increase in an MHCII[+]PBM subpopulation in PS19 mice compared with WT littermates. Consequently, we asked whether PBM dysfunction could exacerbate tau pathology and tau-mediated neurodegeneration. Pharmacological depletion of PBMs in PS19 mice led to an increase in tau pathology and tau-dependent neurodegeneration, which was independent of gliosis or aquaporin-4 depolarization, essential for the CSF-ISF exchange. Together, our results identify PBMs as novel cellular regulators of tau pathology and tau-mediated neurodegeneration.

## Introduction

Alzheimer's disease (AD) is the leading cause of dementia worldwide and is the sixth leading cause of death in the United States. The histopathological hallmarks of AD are extracellular amyloid plaques composed predominantly of the amyloid-$\beta$ (A$\beta$) peptide, and neurofibrillary tangles consisting of abnormally aggregated, hyperphosphorylated tau protein within neurons (Castellani et al, 2010; Holtzman et al, 2011; Serrano-Pozo et al, 2011). In the case of tau, it is noteworthy that its accumulation in the limbic cortex and neocortex correlates with synapse loss and cognitive decline in AD and other tauopathies (Masliah et al, 1992; Nelson et al, 2012; Jadhav et al, 2015). A$\beta$ and tau aggregations in the AD brain result in part from their impaired elimination from the brain to the periphery (Xin et al, 2018).

The recent discovery of meningeal lymphatics highlighted the ability of the brain to drain its debris to deep cervical lymph nodes (Louveau et al, 2015). Indeed, ablation of meningeal lymphatics with either genetic, pharmacological or surgical manipulation resulted in significantly less clearance of injected macromolecules in the parenchyma (Louveau et al, 2015, 2018) and exacerbation of amyloid-$\beta$ pathology in transgenic mice that develop amyloid-$\beta$ deposition (Da Mesquita et al, 2018, 2021b). Importantly, studies have shown that cerebrospinal fluid–interstitial fluid (CSF-ISF) exchange allows A$\beta$ clearance in mouse models of amyloidosis (Iliff et al, 2012; Xu et al, 2015; Abe et al, 2020; Feng et al, 2020). Regarding tau pathology, we previously demonstrated that extracellular tau clearance is impaired in the absence of functional lymphatics (Patel et al, 2019) and other groups showed that the CSF–ISF exchange was altered in a tau mouse model (Harrison et al, 2020; Ishida et al, 2022). Moreover, impaired CSF–ISF exchange is associated with tau hyperphosphorylation after traumatic brain injury (Iliff et al, 2014).

We recently demonstrated that parenchymal border macrophages (PBMs) regulate CSF flow dynamics and their depletion or dysfunction increased accumulation of parenchymal amyloid plaques in the 5xFAD mouse model of amyloidosis (Drieu et al, 2022). PBMs are composed of at least two subpopulations: an antigen presenting cell subpopulation, positioned mainly along the veins and venules, and a scavenger subpopulation located mostly along the arteries and arterioles (Kierdorf et al, 2019). PBMs are long-lasting cells located in the leptomeninges and perivascular spaces and their developmental progenitor is shared with microglial cells (Utz et al, 2020). A recent study highlighted that leptomeningeal macrophages give rise to perivascular macrophages after development, along development of the subarachnoid space (Masuda et al, 2022). Recent reviews highlighted the implication of leptomeningeal and perivascular macrophages in brain-associated pathologies such as bacterial/viral infections, neurodegenerative diseases (multiple sclerosis, amyloidosis), and neurovascular coupling (Faraco et al, 2017; Kierdorf et al, 2019), but their roles in tau

[1]Department of Pathology and Immunology, Washington University School of Medicine, St. Louis, MO, USA    [2]Department of Neurology, Washington University School of Medicine, St. Louis, MO, USA    [3]Hope Center for Neurological Disorders, Washington University School of Medicine, St. Louis, MO, USA    [4]Knight Alzheimer's Disease Research Center, Washington University School of Medicine, St. Louis, MO, USA    [5]Center for Brain Immunology and Glia, Washington University School of Medicine, St. Louis, MO, USA    [6]Institute of Neurophysiopathology (INP UMR7051), Aix-Marseille University, Marseille, France

Correspondence: maud.gratuze@univ-amu.fr; holtzman@wustl.edu

pathology remain largely unexplored. Leptomeningeal/perivascular macrophage depletion has been shown to ameliorate the neuro-vascular dysfunction induced by Aβ (Park et al, 2017), but also to exacerbate perivascular Aβ deposition in a mouse model of cerebral amyloid angiopathy (Hawkes & McLaurin, 2009). Our recent data demonstrated that leptomeningeal/perivascular macrophages (also termed PBMs) regulate CSF flow dynamics, and PBM dysfunction was associated with normal aging (Drieu et al, 2022). We also demonstrated that PBMs play a key function in limiting parenchymal plaque accumulation in 5xFAD mice (Drieu et al, 2022). However, to our knowledge, the link between PBMs and tau pathology has not been addressed.

In this study, we demonstrate an altered CSF flow in PS19 mice, a mouse model that combines tau aggregation with tau-mediated neurodegeneration, compared with their age-matched WT litter-mates. We report an increase in an MHCII⁺-PBM population in PS19 mice, previously associated with aging (Mrdjen et al, 2018; Van Hove et al, 2019; Drieu et al, 2022). Consequently, we asked whether PBM dysfunction could affect tau pathology and tau-mediated neuro-degeneration. Pharmacological depletion of PBMs in PS19 mice led to an increase in tau pathology and tau-dependent neuro-degeneration independently from changes in microglia, astrocytes or subcellular depolarization of aquaporin-4 (AQP4), essential for the CSF–ISF exchange. Together, our results identify PBMs as novel cellular regulators of tau pathology and tau-mediated neuro-degeneration, which can potentially provide a new therapeutic target in AD and other tauopathies.

# Results

## Altered CSF flow in PS19 mice with tau pathology and neurodegeneration

To determine whether CSF flow is altered by tau pathology and tau-related inflammation and neurodegeneration, we used the PS19 tauopathy mouse model that overexpresses 1N4R hTau containing the P301S mutation that causes a familial form of frontotemporal dementia. By 9.5 mo of age, mice (PS19) develop strong tau hyperphosphorylation and aggregation, neurofibrillary tangle de-position, gliosis, neuronal loss, brain atrophy, and loss of synaptic proteins in specific brain regions including the hippocampus, entorhinal cortex, piriform cortex, and amygdala (Yoshiyama et al, 2007). Although extremely useful to model tau pathology and tau-mediated neurodegeneration, PS19 mice are also well-known for their inter-animal variability, implying more animals are required to adequately power an efficacy study (Woerman et al, 2017). To evaluate CSF flow, we injected mice with fluorescent ovalbumin (OVA) into the cisterna magna (i.c.m.), as previously described (Fig 1A) (Iliff et al, 2012; Da Mesquita et al, 2018; Drieu et al, 2022). 1 h after injection, mice were perfused and their brains harvested to assess OVA coverage at the middle cerebral artery level (Fig 1B). We found that PS19 mice showed a reduction of OVA coverage compared with their WT littermates (Fig 1C). Brains were then sectioned and OVA coverage was measured in coronal brain sections. Again, we ob-served a reduction of OVA coverage in PS19 mice (Fig 1D and E).

PBMs have been found to regulate CSF flow dynamics (Drieu et al, 2022). Notably, changes in PBM phenotypes were associated with impaired CSF flow in aged mice. To characterize PBM phenotypes in PS19 mice, we used flow cytometry in whole brains (choroid plexuses were removed; Figs 1F and S1A) and found that the per-centage and total number of PBMs of the CD45⁺ cells were not different in PS19 mice compared with their WT littermates (Figs 1G and S1B). However, PBM phenotypes in PS19 mice were found to be more polarized towards MHCII, indicating that PBMs in PS19 mice were more pro-inflammatory (Figs 1G and S1B).

Collectively, these data showed that PBM dysfunction is asso-ciated with impaired CSF flow in PS19 mice, and that a switch towards inflammatory PBM phenotype is associated with tau pa-thology and tau-mediated neurodegeneration.

## PBM depletion exacerbates tau pathology

To investigate the role of PBMs on the evolution of tau pathology, we performed PBM depletion in PS19 mice using i.c.m. injections of clodronate- (CLO) or PBS- encapsulating liposomes (as control) at 7 and 8.25 mo of age and tissue was analyzed at 9.5 mo of age (Fig 2A). Importantly, ~68% depletion of PBMs was achieved when the brain tissue was examined 1 wk after clodronate injection in 8.25-mo-old PS19 mice (Fig S2A–C). Interestingly, vessel-associated CD206+PBM numbers remained decreased in the hippocampus in 9.5-mo-old CLO-injected PS19 mice (CLO) compared with PBS-injected PS19 mice (CTL) (Fig 2B and C).

Regarding tau pathology, in 9.5-mo-old PS19 mice, when sub-stantial tau pathology had developed, there was no significant difference observed in hippocampus and piriform/entorhinal cortex p-tau staining on serine 202/threonine 205 between PBS- and CLO-treated PS19 mice (Fig 2D–H). However, we observed a significant increase in p-tau staining on serine 409 on both hip-pocampus and piriform/entorhinal cortex (Fig 2I and J), and MC1 staining in hippocampus, revealing an increase in pathological conformational change of tau in PBM-depleted compared with control PS19 mice (Fig 2F and K). Altogether these results suggest that PBM depletion affects more advanced stages of tau pathology. Moreover, we demonstrated a significant positive correlation be-tween hippocampal pathological tau staining and the volume of vessel-associated CD206+PBM in PBM-depleted PS19 mice but not in control PS19 mice, suggesting an association between PBM depletion and increased tau pathology in the hippocampus (Fig S3C and D).

Biochemical analysis of p-tau in hippocampal brain tissue samples by ELISA revealed an increase in both insoluble and soluble p-tau levels (Fig 2M and N), and an increase of insoluble total tau level in PBM-depleted PS19 mice compared with control PS19 mice (Fig 2O). Soluble total tau levels remain unchanged between PBM-depleted and control PS19 mice (Fig 2P).

In summary, these findings demonstrate that PBMs are essential to decelerate the progression of tau pathology in PS19 mice.

## PBM depletion exacerbates tau-mediated neurodegeneration

Because pathological tau is directly linked to neurodegeneration, we next evaluated regional brain volumes in PS19 mice treated for

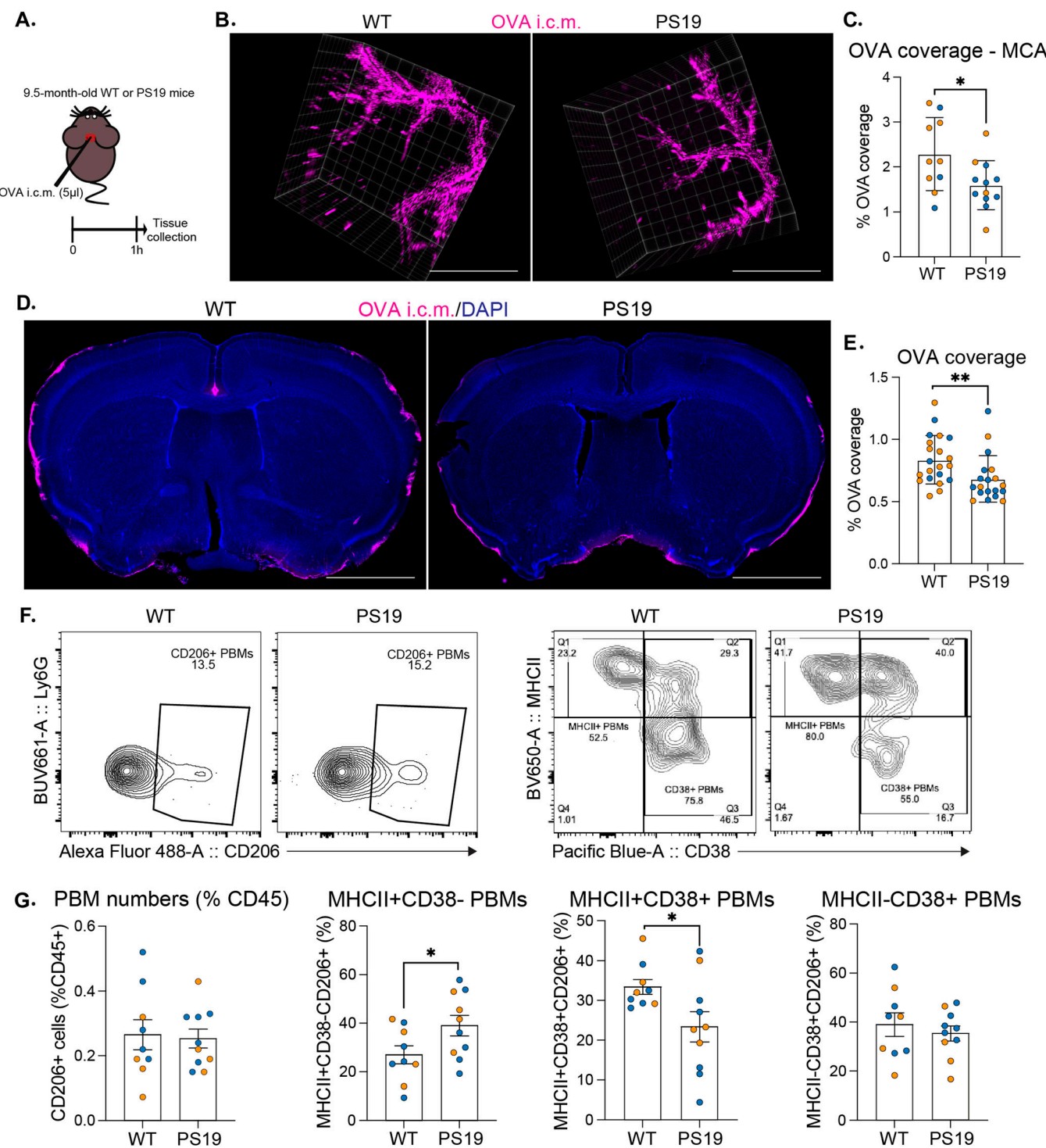

**Figure 1. CSF flow and PBMs in PS19 mice with tau pathology and neurodegeneration.**
**(A)** Experimental schematic: 9.5-mo-old PS19 and WT mice received an i.c.m. injection of OVA; brains were then harvested and imaged 1 h after OVA injection.
**(B)** Representative images showing OVA distribution at the middle cerebral artery (MCA) level. Scale bar, 2 mm. **(C)** Quantification of OVA coverage at the middle cerebral artery level (MCA). n = 10–12 mice/group. **(D)** Representative images showing OVA coverage on brain coronal sections and corresponding quantification of OVA coverage on full-brain coronal sections. Scale bar, 2 mm. **(E)** Quantification of OVA coverage on brain coronal sections. n = 14–26 mice/group; **(F)** Representative contour plots used for the identification of CD206+ cells, and MHCII/CD38 CD206+ subpopulations from half brains of 9.5-mo-old PS19 and WT mice. **(G)** Quantification of CD206+ cells, and subtypes of CD38+MHCII+CD206+, CD38-MHCII+CD206+, and CD38+MHCII-CD206+ cells. n = 9–10 mice/group. Data are presented as mean ± SEM. Significance was determined using two-tailed Mann–Whitney test. *P < 0.05, **P < 0.01, ***P < 0.001 and ****P < 0.0001. Males and females are represented with blue dots and orange dots, respectively.

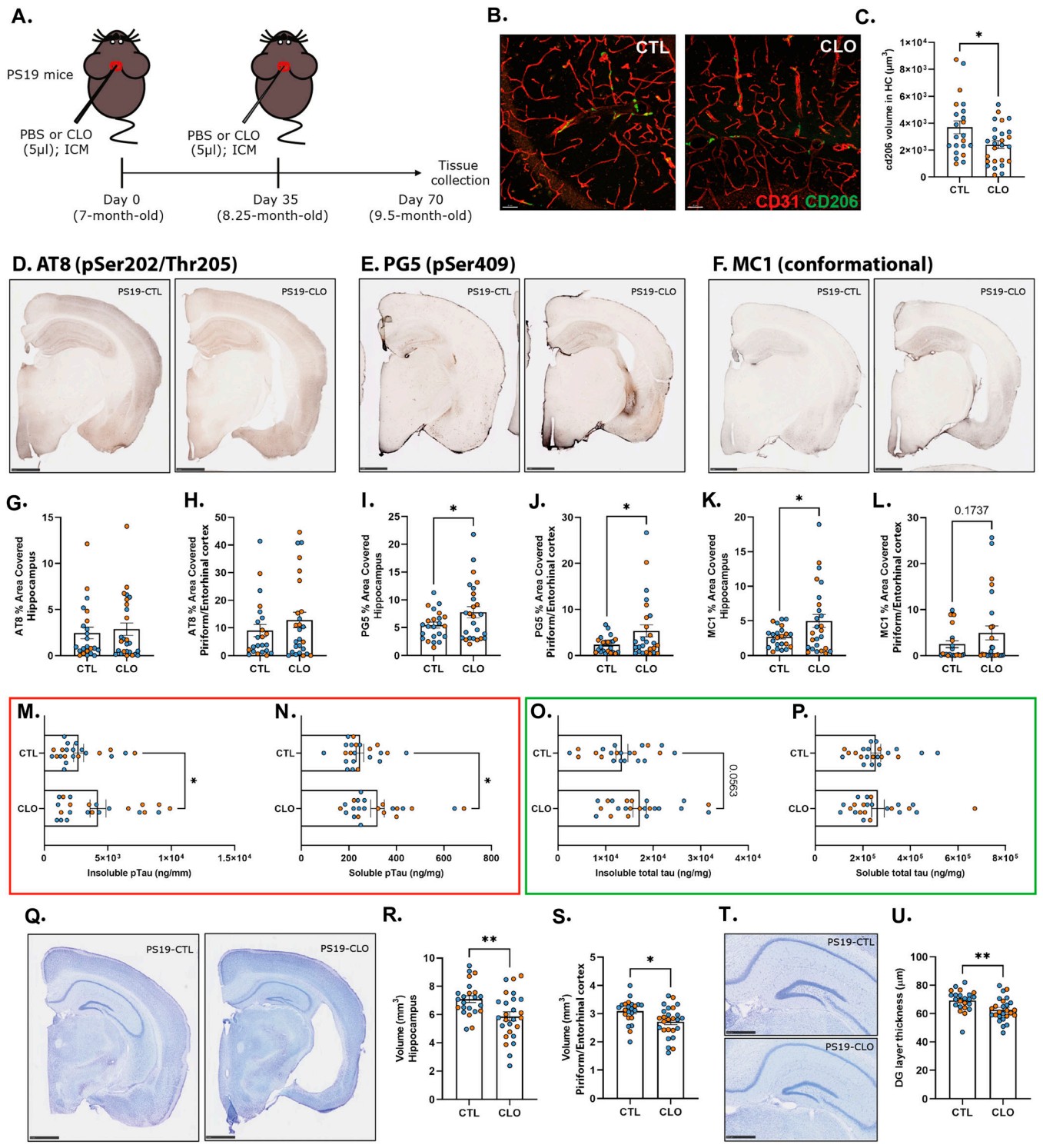

**Figure 2. Pharmacological PBM depletion significantly increased tau pathology and tau-mediated neurodegeneration in PS19 mice.**
**(A)** Experimental schematic: PS19 mice received an i.c.m. injection of clodronate liposomes for PBM depletion or PBS control at 7 and 8.25 mo of age; brains were then harvested at 9.5 mo of age. **(B)** Representative images showing brain sections stained with anti-CD206 (green) and anti-CD31 (red) in the hippocampus of PBM-depleted and control 9.5-mo-old PS19 mice. Scale bar: 50 $\mu$m. **(C)** Quantification of the volume of CD206 cells in close association with CD31$^+$ vessels per image in the CA1 and CA3 regions of the hippocampus. **(D, E, F)** Representative images of p-tau staining ((D): AT8 and (E): PG5) and pathological conformation of tau ((F): MC1) from PBM-depleted and control 9.5-mo-old PS19 mice. Scale bars: 1 mm. **(G, H, I, J, K, L)** Quantification of the percentage area covered by p-tau and pathological tau staining in the hippocampus (G, I, K) and the piriform/entorhinal cortex (H, J, L). **(M, N, O, P)** Concentrations of insoluble (M) and soluble (N) p-tau (p.Ser202/p.Thr205 and p.Thr181) and insoluble (O) and soluble (P) total tau from the hippocampus using a human tau-specific (hTau-specific) sandwich ELISA. **(Q)** Representative images of Nissl staining from 9.5-mo-old PS19 mice. **(R, S)** Quantification of the average volume of the hippocampus (R) and piriform/entorhinal cortex (S). **(T)** Representative images of Nissl

PBS or CLO liposomes. At 9.5 mo of age, we found that PBM depletion exacerbated brain atrophy in PS19 mice compared with control mice (Fig 2Q–S). More specifically, atrophy was significantly increased in the hippocampus (≈+17%; Fig 2R) and in the piriform/entorhinal cortex (≈+12%; Fig 2S). Interestingly, tau pathology inversely correlated with hippocampal volume in PBM-depleted PS19 mice but not in controls (Fig S3A and B) and no direct correlation between PBM numbers and hippocampal atrophy was found (Fig S3E). Then, we estimated neuronal loss by measuring the thickness of the granule cell layer in the dentate gyrus that has been shown to correlate with hippocampal atrophy and neurodegeneration in this model (Shi et al, 2017). In accordance with changes observed in brain volume, the granule cell layer in the dentate gyrus was significantly thinner in PBM-depleted PS19 mice compared with control PS19 mice (Fig 2T and U). This exacerbated neurodegeneration suggests an important role of PBMs in regulating tau-mediated neurodegeneration.

### Gliosis and aquaporin-4 subcellular location remains unaffected by PBM depletion

We and others previously described that microglia and astrocytes strongly contribute to tau pathology and tau-mediated neurodegeneration (Ceyzériat et al, 2018; Mancuso et al, 2019; Shi et al, 2019; Gratuze et al, 2020; Wang et al, 2021a). We next evaluated whether the effects of PBMs on tau pathology and tau-mediated neurodegeneration is associated with modulation of astrocyte and microglial reactivity. We assessed brain sections with the myeloid immune cell marker ionized calcium-binding adaptor molecule 1 (Iba1) to assess the overall microglial population. We also used P2ry12, a homeostatic microglial marker, and Clec7a, a marker of reactive microglia (Fig 3A–D). Interestingly, we observed no difference in the volumes of Iba1 (Fig 3B), Clec7a (Fig 3C), and P2ry12 (Fig 3D) between PBM-depleted and control-treated PS19 mice, suggesting that the effect of PBM depletion on tau pathology and neurodegeneration is independent from microgliosis.

We next focused on astrocytes, known to become reactive in the presence of tau pathology. However, we did not observe any change in GFAP volume, a marker of reactive astrocytes (Fig 3A and E). To further explore whether the effects of PBMs on tau pathology and tau-mediated neurodegeneration are mediated by astrocyte and microglial reactivity, we next assessed mRNA-expression profiles in the hippocampus of several genes associated with disease-associated microglia, astrocyte reactivity, and inflammation (Fig 3F). Over the 60 genes assessed, we observed changes in mRNA levels between PBM-depleted and control-treated PS19 mice for only three genes. Indeed, *abca1*, *apoe*, and *aldh1l1* are up-regulated in PBM-depleted PS19 mice compared with controls, which might reflect changes in cholesterol transport in PBM-depleted mice.

Altogether, our data clearly support that the role of PBM on tau pathology and tau-mediated neurodegeneration is independent of reactive gliosis in the brain parenchyma.

We also studied aquaporin-4 (AQP4), a water channel present in the astrocytic end feet at the outer layer of the perivascular space. AQP4 is a mediator of CSF influx that has been suggested to be depolarized from the perivascular space in AD patients (Ringstad et al, 2017). Furthermore, AQP4-deficient mice showed increased A$\beta$ and tau pathology and neurodegeneration in mice models of AD (Iliff et al, 2012, 2014; Xu et al, 2015; Abe et al, 2020; Harrison et al, 2020; Ishida et al, 2022). Indeed, we observed an increased AQP4 depolarization in PS19 mice compared with WT littermate mice (Fig 4A and B) as previously described in another mouse model of tau pathology (Harrison et al, 2020). However, we did not find any differences in AQP4 depolarization between PBM-depleted and control-treated PS19 mice (Fig 4C). These data support our previous findings suggesting that PBMs modulate CSF flow dynamics independently from AQP4 (Drieu et al, 2022), and that impaired tau clearance leading to increased tau pathology and neurodegeneration could come from a combination between AQP4 depolarization and PBM dysfunction.

## Discussion

Although the importance of microglia or astrocytes on AD pathophysiology has been extensively demonstrated, immune cells at the CNS borders also appear to be involved in AD (Da Mesquita et al, 2021a). We recently demonstrated that PBMs provide a clearance pathway for amyloid-$\beta$ peptide (Drieu et al, 2022). Here, we examined the role of PBMs in the context of tau pathology and tau-mediated neurodegeneration. We demonstrated an altered CSF flow in the presence of tau pathology, associated with an increase of an MHCII⁺PBM subpopulation, previously associated with impaired CSF flow in aging (Drieu et al, 2022), suggesting that tau pathology can promote PBM dysfunction. On the other hand, pharmacological depletion of PBMs in PS19 mice led to an increase of tau pathology and tau-dependent neurodegeneration independently from astrocytes or microglia. Interestingly, the role of PBMs in the context of tau pathology and tau-mediated neurodegeneration seems to be sex-independent in our PS19 model (Table S1). Although it remains unknown whether PBM dysfunction or tau pathology is the primary event, the presence of tau pathology and neurodegeneration along with PBM reactivity is associated with a detrimental circle ultimately leading to CSF flow impairment and exacerbated neurodegeneration. However, the exact mechanisms of how PBM dysfunction led to the exacerbation of tau pathology and neurodegeneration require further investigations.

staining from the dentate gyrus layer of 9.5-mo-old PS19 mice. **(U)** Quantification of the granule cell layer of the dentate gyrus; n = 23–26 mice/group; data are presented as mean ± SEM. Significance was determined by an unpaired, two-tailed *t* test for (C, I, J, L, M, N, O, P, R, S, U). For (G), an unpaired, two-tailed Mann–Whitney test was used because of the nonparametric dataset. For (H, K), two-tailed *t* test with Welch's correction was used because of significantly different variances. *$P$ < 0.05, **$P$ < 0.01, ***$P$ < 0.001, and ****$P$ < 0.0001. Males and females are represented with blue dots and orange dots, respectively.

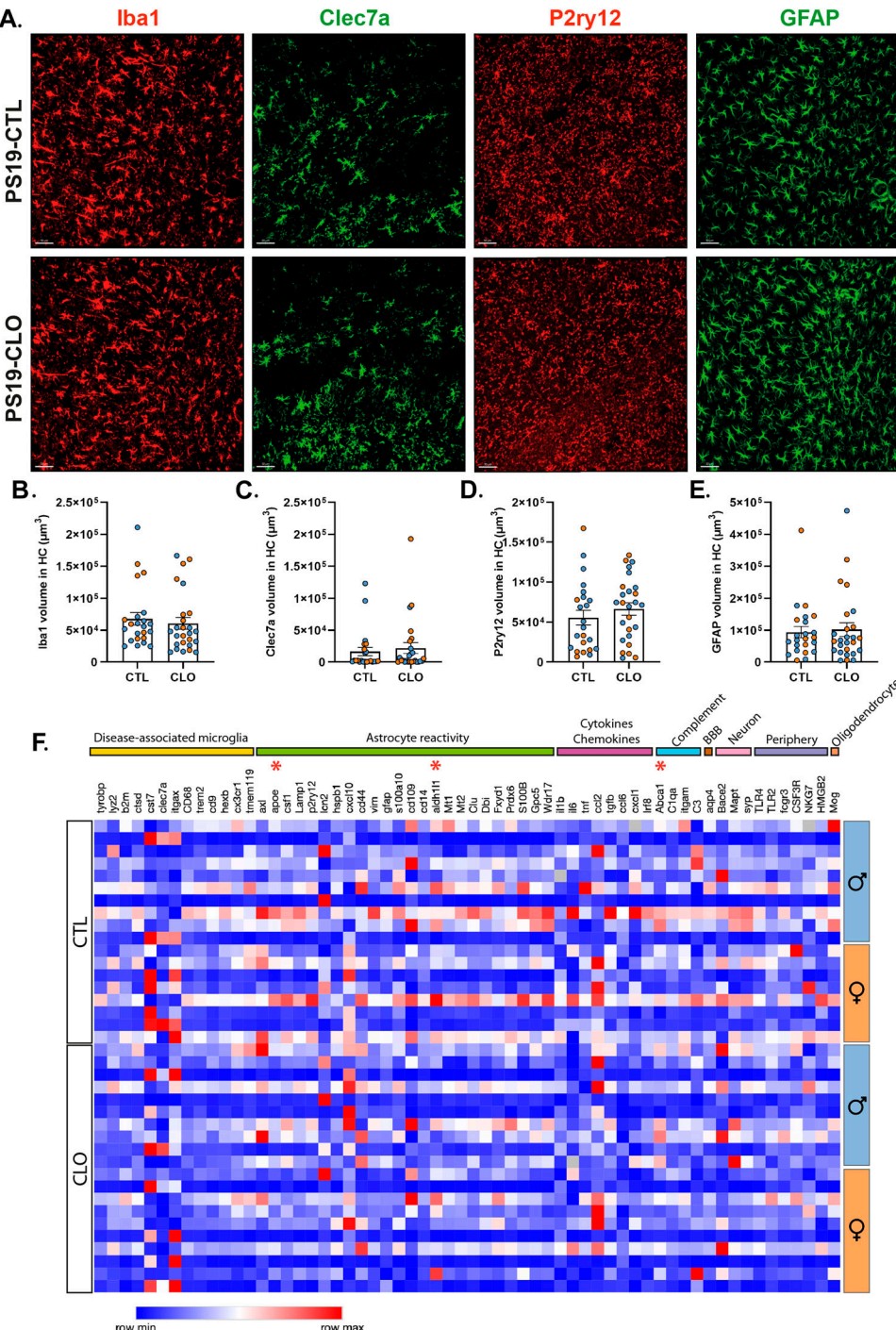

**A.**

PS19-CTL / PS19-CLO

Iba1 | Clec7a | P2ry12 | GFAP

**Figure 3. Pharmacological PBM depletion does not affect gliosis in 9.5-mo-old PS19 mice.**

**(A)** Representative images showing brain sections stained with anti-Iba1 (red), anti-Clec7a (green), anti-P2ry12 (red), and anti-GFAP (GFAP) in the hippocampus of PBM-depleted and control 9.5-mo-old PS19 mice. Scale bar: 50 $\mu$m. **(B, C, D, E)** Quantification of the volume of Iba1[+] (B), Clec7a[+] (C), P2ry12[+] (D), and GFAP[+] (E) cells per image in the CA1 and CA3 regions of the hippocampus n = 23–26 mice/group. **(F)** Heatmap analysis of bulk RNA in the hippocampus of PBM-depleted and control 9.5-mo-old PS19 mice generated by hierarchical gene clustering based on groups. n = 18–20 mice/group. Data are presented as mean ± SEM. For (B, C, D, E, F) significance was determined by an unpaired, two-tailed $t$ test. *$P < 0.05$. Males and females are represented with blue dots and orange dots, respectively.

One possible explanation for the exacerbated tau pathology in PBM-deficient PS19 mice would be that the CSF flow around the arterioles is decreased due to PBM depletion (Drieu et al, 2022), which could increase tau in CSF or ISF and therefore, tau aggregation. In addition, altered CSF flow in PBM-depleted PS19 mice might also exacerbate tau-associated ventricular enlargement and consequently worsen brain atrophy. Alternatively, we previously described that the perivascular space is smaller in PBM-depleted mice than in control mice

(Drieu et al, 2022) and a recent study suggests that enlarged perivascular spaces are associated with decreased brain tau deposition (Kang et al, 2022), which could explain increased tau pathology in the absence of PBM in PS19 mice. In another study, lower perivascular space volume fraction in the medial temporal lobe of MCI patients was not associated with amyloid-PET uptake, but was independently associated with Tau-PET uptake (Sepehrband et al, 2021). However, those hypotheses will need deeper investigation as other studies

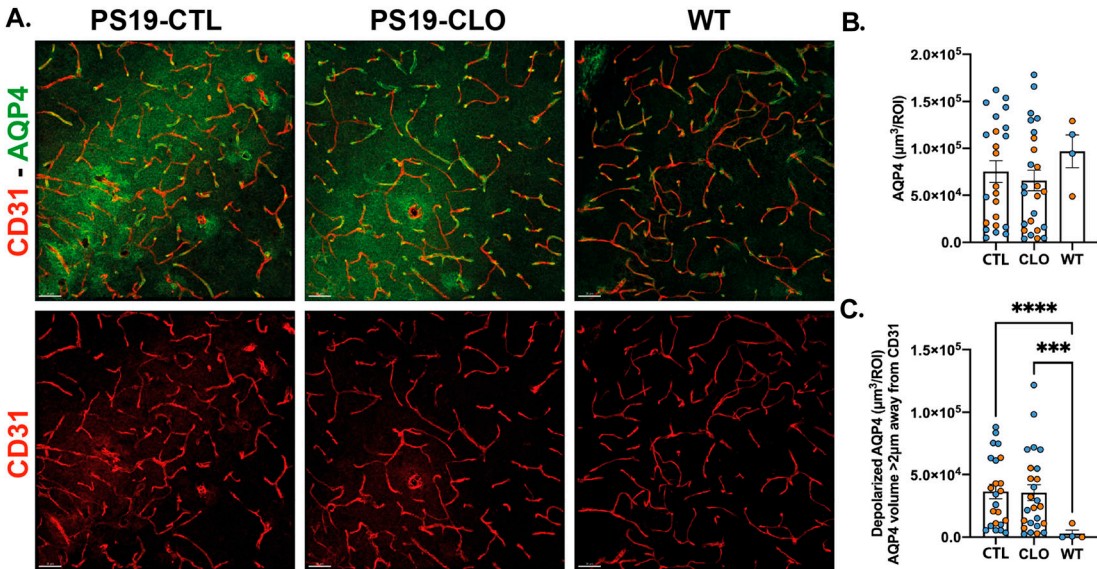

**Figure 4.   Pharmacological PBM depletion does not impact aquaporin-4 polarization in 9.5-mo-old PS19 mice.**
**(A)** Representative images showing brain sections stained with anti-AQP4 (green) and anti-CD31 (red) in the hippocampus of PBM-depleted and control 9.5-mo-old PS19 mice and age-matched WT littermates. Scale bar: 50 $\mu$m. **(B, C)** Quantification of the total volume of AQP4 (B) and the volume of depolarized AQP4 (C) measured as AQP4 volume at least 2 $\mu$m away from CD31$^+$ vessels per image in the CA1 and CA3 regions of the hippocampus. n = 23–26 mice/group (WT n = 4). Data are presented as mean ± SEM. Significance was determined using a one-way ANOVA followed by (A) Tukey's post hoc test for (B) and Welch's and Brown–Forsythe ANOVA tests were used for (C) because of significantly different variances. ***$P$ < 0.001 and ****$P$ < 0.0001. Males and females are represented with blue dots and orange dots, respectively.

suggest that enlarged perivascular space are associated with tau pathophysiology (Wang et al, 2021b; Vilor-Tejedor et al, 2021). Furthermore, a study showed that PS19 mice harbor inflamed brain vasculature (Propson et al, 2021), which could in turn affect blood–brain barrier function/permeability, and consequently CSF clearance. The choroid plexus might also play a role in tau-mediated neurodegeneration, as we and others observed an enlargement of brain ventricles in PS19 mice compared with their control littermates (Yoshiyama et al, 2007; Chen et al, 2023). However, the results of PBM depletion on tau-mediated neurodegeneration appear to be independent on the choroid plexus function, as our depletion model does not target choroid plexus macrophages (Drieu et al, 2022). Investigating the underlying mechanisms will be essential to better understand of PBM and tau pathology impact on each other, ultimately leading to exacerbated neurodegeneration.

Together, our results identify PBM as novel cellular regulators of tau pathology and tau-mediated neurodegeneration, which can potentially provide a new therapeutic target in AD and other tauopathies.

# Materials and Methods

### Mice

PS19 tau transgenic mice (Stock No. 008169; Jackson Laboratories) harbors 1N4R tau overexpressing the human P301S tau mutation, driven by the mouse prion protein PrP promoter. These mice have been backcrossed to C57BL/6 mice (Stock No.

027; Charles River) three times. All PS19 transgenic and WT mice littermates involved in the final analyses were obtained from the same generation. Littermates of the same sex were randomly assigned to experimental groups. Both males and females were used in this study. All mice were housed under a normal 12-h light/dark cycle in a temperature- and humidity-controlled environment with water and food ad libitum. All animal procedures and experiments were performed under guidelines approved by the Institutional Animal Care and Use Committee (IACUC) at Washington University School of Medicine. All experiments and data analysis were accomplished by researchers who were blind to genotype and treatments.

### I.C.M. injections

Mice were anesthetized using an intraperitoneal injection of a cocktail of ketamine (100 mg/kg) and xylazine (10 mg/kg) (KX) diluted in 0.9% saline solution. The fur of the neck was shaved and cleaned with 70% iodine. Mice were placed in a stereotaxic frame to maintain the head fixed, and an ophthalmic solution was applied to prevent drying eyes. The skin from the neck was longitudinally incised and muscles were retracted using hooks to expose the cisterna magna. The solutions, diluted in artificial CSF (aCSF), were injected using a 33-gauge Hamilton syringe (5 $\mu$l; 2.5 $\mu$l/min). The syringe was left in place for 1 min after injection to prevent backflow. At the end of the injection, the skin was sutured, and mice were kept on a heating pad until fully awake. Mice received a subcutaneous injection of ketoprofen (2.5 mg/kg) at the end of the surgery.

## I.C.M. OVA injection

Alexa647-conjugated OVA (45 kD; 5 µl; 1 mg/ml diluted in artificial CSF) was i.c.m.-injected to evaluate CSF flow dynamics and mice were perfused 1 h later.

## PBM depletion

To deplete PBMs, PS19 mice received two i.c.m. injections of clodronate-loaded liposomes (CLO; 5 µl; 5 mg/ml; CLD-8901; Thermo Fisher Scientific) at 7 and 8.25 mo of age. The control PS19 group received two i.c.m. injections of PBS-loaded liposomes (PBS) at 7 and 8.25 mo of age. To evaluate the efficiency of CLO–liposomes, 8.25 mo of age PS19 mice received an i.c.m. injection of PBS– or CLO–liposomes, and quantification of PBM numbers was made 1 wk later (Fig S2).

## Tissue collection

For the CSF flow experiment, 9.5-mo-old PS19 and littermates WT mice received a lethal intraperitoneal injection of euthasol (10% vol/vol in saline, 250 µl) and transcardially perfused with PBS containing 10 U/ml heparin. After removal of the skin, muscles, and mandibles, the head was drop-fixed in 4% PFA for 24 h. Then, the skull caps (skull and attached dorsal dura mater) were detached and brains were kept in 4% PFA for an additional 24 h (48 h in total) postfixation. The tissues were then cryoprotected with 30% sucrose solution and frozen in Tissue-Plus OCT compound (Thermo Fisher Scientific). Brains were sliced (40- or 100-µm-thick sections) with a cryostat and kept in 24-well plates filled with PBS at 4°C.

For the PBM-depletion experiment, CLO- or PBS-injected PS19 mice were euthanized 9.5 mo of age by intraperitoneal injection of pentobarbital (200 mg/kg). Blood samples were collected in EDTA-treated tubes before cardiac perfusion with 3 U/ml heparin in cold Dulbecco's PBS. Blood samples were spun down (10 min, 2,000*g*, 4°C), and blood plasma was collected. Brains were carefully extracted and cut into two hemispheres. The left hemisphere was collected for immunostaining and immerse-fixed in 4% PFA overnight before being transferring to 30% sucrose and stored at 4°C until sectioned. Brains were cut coronally into 50-µm sections on a freezing sliding microtome (HM 400; Microm) and stored in cryoprotectant solution (0.2 M PBS, 15% sucrose, 33% ethylene glycol) at −20°C until use. The right hemisphere was dissected to isolate the hippocampus and the cortex for biochemical analysis, and the tissue was kept at −80°C until analyzed.

For the flow cytometry experiment, mice were perfused using PBS containing 10 U/ml heparin. Brains were then extracted and separated in two hemispheres and incubated in FACS buffer. Cerebellum, olfactory bulbs, and lateral choroid plexuses were removed.

## FACS

Half-brains from 9.5-mo-old PS19 and littermates WT mice were gently dissected using scissors and incubated in digestion solution three times for 20 min each (brains were also mechanically grinded using descending diameter plastic pipettes between incubations). Tissues were then mashed through 70-µm strainers in 50 ml tubes containing FACS buffer and 10% FBS to stop enzymatic digestion. Myelin was removed by transferring samples into 3 ml FACS buffer containing 22% BSA and centrifuged (1,000*g*; nine accelerations; two decelerations; 10 min at 4°C). The remaining supernatant and the myelin layer were carefully removed, and the pellet was resuspended in FACS buffer and transferred to a V-bottom plate. 50 µl of FcBlock solution (1:50) were added to the wells and after 20 min. The following antibodies (1:200 dilution; eBioscience) were used in this study: rat anti-mouse Ly6C (128033, BV510; BioLegend), rat anti-mouse Ly6G (741587, BUV661; BD Biosciences), rat anti-mouse CD45 (746947, BV750; BD Biosciences), rat anti-mouse CD19 (751213, BUV615; BD Biosciences), rat anti-mouse CD11b (741242, BUV563; BD Biosciences), rat anti-mouse TCRb (748405, BUV805; BD Biosciences), rat anti-mouse F4/80 (123133, BV605; BioLegend), rat anti-mouse CD64 (139308, PerCP-Cy5.5; BD Biosciences), rat anti-mouse CD206 (141710, AF488; BioLegend), rat anti-mouse CD38 (102716, AF647; BioLegend or 102719, Pacific Blue; BioLegend), rat anti-mouse MHCII (107641, BV650; BioLegend). Cell viability was determined using DAPI staining. Fluorescence data were acquired using Cytek Aurora spectral flow cytometer (Cytek) then analyzed using FlowJo Software (Tree Star, v5.0).

## Immunohistochemistry and immunofluorescence

For immunohistochemical p-tau staining (AT8-biotinylated, mouse monoclonal, 1:500, MN1020B—PG5 and MC1, mouse monoclonal, 1:500; Thermo Fisher Scientific [gift from Peter Davies]), three sections were washed in TBS buffer three times, for 5 min each. After washing, sections were incubated in 0.3% hydrogen peroxide in TBS for 10 min at room temperature to block endogenous peroxidase activity, followed by three washes in TBS, for 5 min each. Sections were then blocked with 3% milk in TBS with 0.25% Triton X-100 (TBSX) for 1 h at room temperature to prevent nonspecific antibody binding. Then, the sections were incubated at 4°C overnight with primary antibodies. The next day, after three washes in TBS, the sections incubated with PG5 or MC1 antibodies were incubated with goat anti-mouse HRP-conjugated secondary antibody (1:500) in 1% milk in TBS with 0.25% Triton X-100 for 1 h at room temperature. For AT8 staining, after washing, sections were incubated in ABC Elite solution (PK-6100; VectaStain) for 1 h, followed by three times washing in TBS. Finally, the sections were developed and stained using DAB Peroxidase Substrate, washed, and mounted on slides. After drying overnight, the slides were dehydrated in increasing ethanol concentrations followed by xylene and coverslipped with Cytoseal 60.

For immunofluorescent staining of GFAP (mouse monoclonal GFAP Alexa Fluor 488, 1:500, 53-9892-82; Invitrogen), Iba1 (rabbit polyclonal, 1:2,000, 019-19741; FUJIFILM Wako), P2ry12 (rabbit polyclonal, 1:500, gift from Oleg Butovsky), Clec7a (rat monoclonal, 1:50, mabg-mdect; InvivoGen), CD206 (rat monoclonal, 1:500, MCA2235; Bio-Rad), CD31 (Armenian hamster monoclonal, 1:200, MAB1398Z, clone 2H8; Millipore Sigma), AQP4 (rabbit polyclonal, 1:500, A5971; Millipore Sigma), three sections were washed in PBS three times, for 5 min each. After washing, the sections were permeabilized with 0.25% PBSX for 30 min, followed by three washes in TBS, for 5 min each. Then, the sections were blocked in 3% BSA/3% donkey serum in 0.1% PBSX for 1 h at room temperature, followed by

overnight incubation at 4°C with primary antibodies. The next day, after three washes in PBS, sections were incubated with corresponding fluorescence-labeled secondary antibodies (Invitrogen) for 2 h at room temperature. Then, the sections were washed in PBS three times, for 20 min each, followed by incubation in 0.1% Sudan black solution in 70% ethanol for 10 min. Finally, the sections were washed in PBS three times and mounted in a Fluoromount-G slide mounting medium.

### Image acquisition and analysis

Images were obtained from three sections per mouse for IHC and IF. For IHC, slides were scanned on the NanoZoomer 2.0-HT system (Hamamatsu Photonics). Images were further processed using NDP-viewing software (Hamamatsu) and Fiji software version 1.51 (National Institutes of Health). For IF images, six z-stacks per mouse were acquired on a Leica Stellaris confocal (5 or 8) with a 20x objective and 1,024 × 1,024 resolution. Quantification of confocal images were performed on a semi-automated platform using MATLAB and Imaris 9.7.1 software to create surfaces of each stain based on a threshold applied to all images and colocalize surfaces to evaluate the volume of CD206 cells within CD31$^+$ vessels.

Quantification of confocal images for AQP4 depolarization from CD31$^+$ vessels was performed on a semi-automated platform using MATLAB and Imaris 9.7.1 software (Bitplane) to create surfaces of each stain based on a threshold applied to all images, dilate CD31 surfaces to 2 μm, and analyses AQP4 volume outside the CD31-dilated surface.

For brain tracer-coverage measurements, six sections per mouse were used and manually thresholded to match the observed signals. All six images were quantified as area of signal/total area (DAPI coverage) and averaged to get one value per mouse.

### Volumetric analysis of brains and neuronal layer thickness measurement

Brain sections were mounted and allowed to dry overnight. The following day, sections were stained in cresyl violet for 5 min at room temperature. The slices were then sequentially dehydrated in increasing ethanol concentrations followed by xylene and coverslipped with Cytoseal 60 (8310-16; Thermo Fisher Scientific). Slides were scanned using Hamamatsu's Nanozoomer microscope at 20X magnification. Hippocampus and entorhinal/piriform cortex were traced using the NDP view 2. Volumetric analysis of the hippocampus and the entorhinal/piriform cortex was performed via stereological methods by assessing sections spaced 300 μm starting from bregma –1.2 mm to bregma –3.3 mm (6–8 sections per mouse depending on the severity of brain atrophy). The formula for the volumetric calculation was volume = (sum of area) ∗ 0.3 mm. Three sections, corresponding approximately to bregma coordinates –2.1, –2.4, and –2.7 mm were selected to measure the thickness of the dentate gyrus granular cell layer by drawing a scale perpendicular to the cell layer in all three slices and taking the average value for each mouse.

### Soluble/insoluble tau extraction & tau and p-tau ELISA

Half of the hippocampus was weighed and homogenized using a pestle with 20 μl buffer/mg of tissue (10 mM Tris–HCl pH 7.4, 0.8 M NaCl, 1 mM EDTA, 2 μM DTT, cOmplete and PhosStop [both Roche], and 10% sucrose). Samples were centrifuged for 10 min at 10,000$g$ and 4°C. Supernatant was removed and kept on ice, whereas the pellet was re-homogenized in the same volume of buffer with a sonicator at 30% amplitude, 1 s on/1 s off pulse, for 1 min, and centrifuged for 10 min at 10,000$g$ and 4°C. The two supernatants were pooled together and incubated with 1% Sarkosyl rotating at RT for 1 h. Samples were then centrifuged for 1 h at 300,000$g$ and 4°C. The supernatant is the sarkosyl-soluble fraction (SS). The pellet was washed in PBS, suspended in PBS and sonicated at 30% amplitude, 1 s/1 s pulse, for 1 min (Sarkosyl-insoluble fraction [SI]). SS and SI fractions were frozen until used for ELISA or Western Blot. The concentration of tau and p-tau was quantified by sandwich ELISA using Tau-5 (in-house antibody) as the coating antibody and human-specific biotinylated HT7 for detection for tau ELISA and using HJ14.5 (in-house p.Thr181-tau antibody) as the coating antibody and human-specific biotinylated AT8 for detection for p-tau ELISA. Briefly, 96-well half-area plates were coated with 2 μg/ml of either HJ14.5 or Tau-5 antibody and incubated at 4°C overnight. The next day, the plate was blocked in 3% BSA (RPI Corp.) in PBS for 1 h at 37°C. Next, peptide standards and samples were diluted in the sample buffer (0.25% BSA/PBS, 1×protease inhibitor, 300 mM Tris pH 7.4, PBS), loaded onto the plate, and incubated at 4°C overnight. On the 3rd d, 0.3 μg/ml of biotinylated AT8 for p-tau ELISA (MN1020B; Thermo Fisher Scientific) or biotinylated HT7 for tau ELISA (MN1000B; Thermo Fisher Scientific) was applied to the plate for 1.5 h at 37°C, and then Streptavidin–poly-HRP-40 (1:10,000 for tau and 1:6,000 for p-tau; Fitzgerald) was applied for 1.5 h at room temperature. TMB Superslow Substrate solution (Millipore Sigma) was added and the plates were read at 650 nm on a BioTek plate reader after developing for 30 min at room temperature. All samples were run in duplicate.

### Fluidigm qRT-PCR

We extracted total RNA from mouse hippocampus with the RNeasy Mini Kit (catalog no. 74104; QIAGEN) and prepared cDNA with the high-capacity RNA-to-cDNA kit (cat# 4388950; Applied Biosystems) following the manufacturer's instructions. cDNA was further purified using QIAquick PCR purification kit following the manufacturer's instructions (cat# 28104; QIAGEN). Gene expression analysis was performed using microarray in collaboration with the Genome Technology Access Core at Washington University. Using TaqMan probes, the relative gene expression was quantitatively measured using Fluidigm Biomark HD with integrated fluidic circuits. The data were normalized by evaluation of geometricCt mean of housekeeping genes (GAPDH, RN18S1 and ACTB) for each sample. Then the expression levels were calculated according to the ΔΔCt method.

### Statistics

All data are presented as mean ± SEM. GraphPad Prism 9.0.0 was used to perform statistical analyses. Gaussian distribution was

evaluated using the D'Agostino–Pearson normality test. Statistical analyses were performed using two-tailed unpaired $t$ test under normal distribution. In case of unequal variances, Welch's correction was used with the unpaired $t$ test. If samples deviated from normal distribution, statistical analyses were performed using a Mann–Whitney test. Statistical analyses for more than two groups were performed using ordinary one-way ANOVA followed by a Tukey's post hoc test under a normal distribution. In case of unequal variances, Welch's and Brown–Forsythe tests were used. If samples deviated from the normal distribution, statistical analyses were performed using Kruskal–Wallis test followed by a Dunn's post hoc test. $P$-value less than 0.05 ($P < 0.05$) was considered as statistic significant difference. Data from Table S1 were analyzed using two-way ANOVA. *$P < 0.05$; **$P < 0.01$; ***$P < 0.001$; ****$P < 0.0001$.

# Supplementary Information

# Acknowledgements

This work was supported by grants from the National Institutes of Health/National Institute on Aging (AG034113, AG057496, AG078106), the Cure Alzheimer's Fund, and the Ludwig Foundation (to J Kipnis). We thank all the members of the Kipnis and the Holtzman laboratories for their valuable comments during numerous discussions of this work. This study was supported by a grant from the BrightFocus foundation (A2020257F, M Gratuze), NIH grant AG047644, the JPB Foundation, the Cure Alzheimer's Fund, the Charles and Helen Schwab Foundation (to DM Holtzman), and the Edward N and Della L Thome Memorial Foundation, Bank of America, N.A., Trustee (to DM Holtzman). Scanning of immunohistochemistry was performed on the NanoZoomer digital pathology system courtesy of the Hope Center Alafi Neuroimaging Laboratory.

## Author Contributions

A Drieu: conceptualization, data curation, formal analysis, investigation, visualization, methodology, and writing—original draft, review, and editing.
S Du: data curation, formal analysis, investigation, and methodology.
M Kipnis: formal analysis, investigation, and methodology.
ME Bosch: formal analysis, investigation, methodology, and writing—review and editing.
J Herz: data curation, formal analysis, investigation, methodology, and writing—review and editing.
C Lee: investigation and methodology.
H Jiang: investigation and methodology.
M Manis: investigation and methodology.
JD Ulrich: data curation, formal analysis, validation, visualization, and writing—review and editing.
J Kipnis: supervision, funding acquisition, investigation, and writing—original draft, review, and editing.
DM Holtzman: supervision, funding acquisition, investigation, and writing—original draft, review, and editing.
M Gratuze: conceptualization, data curation, formal analysis, validation, investigation, visualization, methodology, and writing—original draft, review, and editing.

## Conflict of Interest Statement

DM Holtzman is listed as an inventor on a provisional patent from Washington University on TREM2 antibodies. DM Holtzman is listed as inventor on a patent licensed by Washington University to C2N Diagnostics on the therapeutic use of anti-tau antibodies. DM Holtzman co-founded and is on the scientific advisory board of C2N Diagnostics. DM Holtzman is on the scientific advisory board of Denali, Genentech, and Cajal Neurosciences and consults for Asteroid Therapeutics.

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
