## [Reviewer comments · Life Science Alliance]

Life Science Alliance

Parenchymal border macrophages regulate tau pathology and tau-mediated neurodegeneration

Antoine Drieu, Siling Du, Michal Kipnis, Megan Bosch, Jasmin Herz, Choonghee Lee, Hong Jiang, Melissa Manis, Jason Ulrich, Jonathan Kipnis, David Holtzman, and Maud Gratuze

DOI: <https://doi.org/10.26508/lsa.202302087>

Corresponding author(s): Maud Gratuze, Aix-Marseille University and David Holtzman, Washington University School of Medicine

Review Timeline:

Submission Date:	2023-04-10
Editorial Decision:	2023-04-18
Revision Received:	2023-07-26
Editorial Decision:	2023-07-28
Revision Received:	2023-07-31
Accepted:	2023-07-31

Transaction Report:

Please note that the manuscript was previously reviewed at another journal and the reports were taken into account in the decision-making process at *Life Science Alliance*. Since the original reviews are not subject to Life Science Alliance's transparent review process policy, the reports and author response cannot be published

April 18, 2023

Re: Life Science Alliance manuscript #LSA-2023-02087-T

Dr. Maud Gratuze
Aix-Marseille University
Marseille
France

Dear Dr. Gratuze,

Thank you for submitting your manuscript entitled "Parenchymal border macrophages regulate tau pathology and tau-mediated neurodegeneration" to Life Science Alliance. We invite you to submit a revised manuscript addressing the Reviewer comments.

Thank you for this interesting contribution to Life Science Alliance. We are looking forward to receiving your revised manuscript.

Sincerely,

B. MANUSCRIPT ORGANIZATION AND FORMATTING:

July 28, 2023

RE: Life Science Alliance Manuscript #LSA-2023-02087-TR

Dr. Maud Gratuze
Aix-Marseille University
27 Boulevard Jean Moulin
Marseille 13005
France

Dear Dr. Gratuze,

Thank you for submitting your revised manuscript entitled "Parenchymal border macrophages regulate tau pathology and tau-mediated neurodegeneration". We would be happy to publish your paper in Life Science Alliance pending final revisions necessary to meet our formatting guidelines.

- please consult our manuscript preparation guidelines <https://www.life-science-alliance.org/manuscript-prep> and make sure your manuscript sections are in the correct order (there are also two Acknowledgement sections)
- please add ORCID ID for corresponding author--you should have received instructions on how to do so
- please add the Twitter handle of your host institute/organization as well as your own or/and one of the authors in our system
- please use the [10 author names, et al.] format in your references (i.e. limit the author names to the first 10)
- please add a callout for Fig S1A, B ; Fig S2A, B, C ; Fig S3A, B, E to your main manuscript text

A. FINAL FILES:

B. MANUSCRIPT ORGANIZATION AND FORMATTING:

**Submission of a paper that does not conform to Life Science Alliance guidelines will delay the acceptance of your

manuscript.**

The license to publish form must be signed before your manuscript can be sent to production. A link to the electronic license to publish form will be sent to the corresponding author only. Please take a moment to check your funder requirements.

Sincerely,

July 31, 2023

RE: Life Science Alliance Manuscript #LSA-2023-02087-TRR

Dr. Maud Gratuze
Aix-Marseille University
27 Boulevard Jean Moulin
Marseille 13005
France

Dear Dr. Gratuze,

Thank you for submitting your Research Article entitled "Parenchymal border macrophages regulate tau pathology and tau-mediated neurodegeneration". It is a pleasure to let you know that your manuscript is now accepted for publication in Life Science Alliance. Congratulations on this interesting work.

DISTRIBUTION OF MATERIALS:

Again, congratulations on a very nice paper. I hope you found the review process to be constructive and are pleased with how the manuscript was handled editorially. We look forward to future exciting submissions from your lab.

Sincerely,
